# Improving Deep Ensembles without Communication

**Konstantinos Pitas**
Statify Team
Inria Grenoble Rhône-Alpes
`pitas.konstantinos@inria.fr`

**Michael Arbel**
Thoth Team
Inria Grenoble Rhône-Alpes
`michael.arbel@inria.fr`

**Julyan Arbel**
Statify Team
Inria Grenoble Rhône-Alpes
`julyan.arbel@inria.fr`

## Abstract

Ensembling has proven to be a powerful technique for boosting model performance, uncertainty estimation, and robustness in supervised deep learning. We propose to improve deep ensembles by optimizing a tighter PAC-Bayesian bound than the most popular ones. Our approach has a number of benefits over previous methods: 1) it requires no communication between ensemble members during training to improve performance and is trivially parallelizable, 2) it results in a simple soft thresholding gradient update that is much simpler than alternatives. Empirically, we outperform competing approaches that try to improve ensembles by encouraging diversity. We report test accuracy gains for MLP, LeNet, and WideResNet architectures, and for a variety of datasets.

## 1 Introduction

Ensembling combines predictions from multiple trained models. In the deep learning setting, it has proven effective at improving model accuracy as well as capturing predictive uncertainty, outperforming Bayesian approaches for the same number of posterior samples [Arbel et al., 2023]. To be effective, each ensemble member has to capture useful features from the data distribution. Usually this is enforced by encouraging the ensemble to be diverse, that is different ensemble members should capture different features.

In standard deep ensembles [Lakshminarayanan et al., 2017], which remain the gold standard for most tasks, each ensemble member is initialized with a different random set of weights. Each ensemble member is then trained independently (typically with standard SGD) and some diversity is induced by chance since each ensemble member converges to a different minimum and the predictive functions of different minima are empirically diverse [Fort et al., 2019]. We propose a method that improves deep ensembles by leveraging the diversity effect of different initializations, while also biasing the minima to have desirable generalization properties. To achieve this, we diverge significantly from the existing literature.

- Existing approaches try to improve ensembles by promoting diversity through continuous communication between the training procedures of the different ensemble members [Ortega et al., 2022, D'Angelo and Fortuin, 2021, Ramé and Cord, 2021, Masegosa, 2020, Wenzel et al., 2020]. Enforcing diversity intuitively requires computing some mean prediction with respect to which the ensemble members are pushed away. This introduces memory costs and

Workshop on Advancing Neural Network Training at 37th Conference on Neural Information Processing Systems (WANT@NeurIPS 2023).

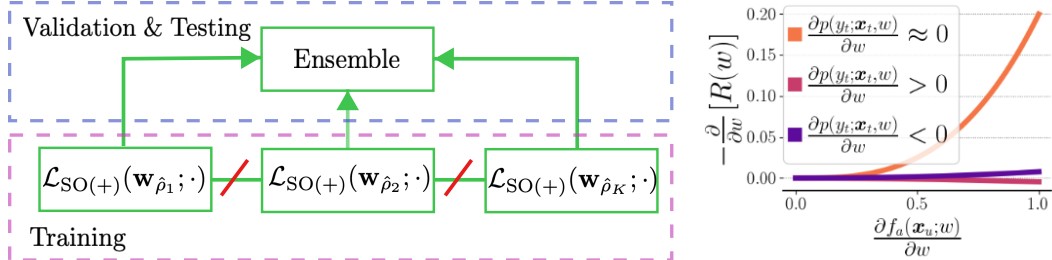

Figure 1: Left: Our approach allows us to train each ensemble member without communication with other ensemble members. We only aggregate the ensemble members for the validation and testing stages. The predictions of each ensemble member on unlabeled data are forced to be different from its predictions on the training set. Our regularisation together with the different initialization of each member encourages the ensemble to be diverse. Right: Value of the regularization term partial derivative for different inputs. Consider a single training $x_t$ and unlabeled $x_u$ sample: non-zero gradient steps, ie $\frac{\partial}{\partial w}[R(\mathbf{w})] \neq 0$, are taken only when $\frac{\partial p(y_t; x_t; \mathbf{w})}{\partial w} \approx 0$ and $\frac{\partial f_a(x_u; \mathbf{w})}{\partial w} \neq 0$. Intuitively, we achieve regularization by fitting all classes or a random class $a$ for each unlabeled sample, *as long as this does not interfere with fitting the labels on the training set.*

> hinders parallelization. Requiring constant communication also complicates the distributed learning setting where one would wish to train each ensemble locally by different agents.

- Existing diversity objectives are hard to optimize. For example, some approaches use adversarial training which is known to be difficult to tune [Ramé and Cord, 2021]. Others push ensembles to be diverse in weight space which interacts non-trivially with function space, the space we are interested in [D'Angelo and Fortuin, 2021]. We believe that such approaches make convergence difficult and thus often fail to materialize gains in test metrics.

We take into account all the above intuitions. We propose a regularization method that does not require communication between different ensemble members to improve test performance, except for the final validation (and testing) step. We encourage each ensemble member to optimize a novel PAC-Bayes bound, which incorporates unlabeled data. Our regularization encourages each ensemble member to generalize well. However, at the same time, we use a different seed for each ensemble member to increase diversity.

Intuitively, our regularizer fits either all classes or a random class for each unlabeled sample, as long as this does not interfere with fitting the labels on the training set. Our approach is a heuristic inspired by a Bayesian interpretation of neural networks. However, we completely avoid injecting additional stochasticity into our training procedure, making our objective easy to optimize. The beneficial properties of our approach, as well as the proposed regulariser, are illustrated in Figure 1. The final ensemble yields consistent improvements over standard ensembles and other approaches.

## 2   Background

A number of approaches have been proposed to improve upon standard deep ensembles.

Wenzel et al. [2020] propose to induce diversity by training on different random initializations as well as different choices of hyperparameters such as the learning rate and the dropout rates in different layers. Ensemble members can be trained independently, and the approach results in consistent gains over standard ensembles. However, the ensemble size increases *quadratically*.

Much closer to our approach are methods that explicitly promote diversity, while retaining the same number of ensemble members. Masegosa [2020], Ortega et al. [2022] propose to optimize a second-order PAC-Bayes bound so as to enforce diversity. In practice, this means estimating the mean log-likelihood of a true label across different ensemble members and "pushing" the different members to estimate a different value for their own likelihood. The authors show improvements for small-scale experiments but cannot improve on large-scale settings. Ramé and Cord [2021] propose to use a discriminator that forces the latent representations of each ensemble member just before the final classification layer to be diverse. They show consistent improvements for large-scale settings in terms

of test accuracy and other metrics, *for the same number of ensemble members*. Yashima et al. [2022] push the latent representations just before the classification layer to be diverse by leveraging Stein Variational Gradient Descent (SVGD). They show improvements in robustness to non-adversarial noise. However, they do not show improvements over Ramé and Cord [2021] in other metrics.

The method closest to our approach is the very recently proposed Agree to Disagree algorithm [Matteo et al., 2023]. This constructs an ensemble greedily by forcing each new member to disagree with previous members on unlabeled data. Crucially, this approach has however been evaluated only on OOD tasks.

The above methods exhibit all the shortcomings we previously described: 1) they require constant communication between ensembles to improve ensembles by promoting diversity, 2) they achieve negligible and/or inconsistent gains which we hypothesize is due to difficult-to-tune training procedures.

One can also approach ensembles as performing approximate Bayesian inference [Wilson and Izmailov, 2020]. One would hope that the regularizing effect of the Bayesian inference procedure would improve the resulting ensembles. Unfortunately, approximate Bayesian inference approaches are typically outperformed by *standard* deep ensembles [Ashukha et al., 2019]. In particular, to achieve the same misclassification or negative log-likelihood error, MCMC approaches typically require many more ensemble members than standard ensembles.

## 3 Second order ensembles

We start our analysis by modeling our predictor using the linearized Laplace approximation [Immer et al., 2021]. This turns the neural network into a Gaussian process with some mean and covariance structure [Immer et al., 2021, Khan et al., 2019]. Instead of optimizing a stochastic objective, we propose to instead enforce the mean and the covariance to have some desired values in appropriate regions of the input space, thus potentially avoiding having to deal with excessive variance in the gradients.

### 3.1 Notations and definitions

We denote the learning sample $(X, Y) = \{(\boldsymbol{x}_i, y_i)\}_{i=1}^n \in (\mathcal{X} \times \mathcal{Y})^n$, that contains $n$ input-output pairs, and use the generic notation $Z$ for an input-output pair $(X, Y)$. Observations $(X, Y)$ are assumed to be sampled randomly from a distribution $\mathcal{D}$. Thus, we denote $(X, Y) \sim \mathcal{D}^n$ the i.i.d observation of $n$ elements. We consider loss functions $\ell : \mathcal{F} \times \mathcal{X} \times \mathcal{Y} \to \mathbb{R}$, where $\mathcal{F}$ is a set of predictors $f : \mathcal{X} \to \mathcal{Y}$. We also denote the risk $\mathcal{L}_{\mathcal{D}}^{\ell}(f) = \mathbf{E}_{(\boldsymbol{x}, y) \sim \mathcal{D}} \ell(f, \boldsymbol{x}, y)$ and the empirical risk $\hat{\mathcal{L}}_{X,Y}^{\ell}(f) = (1/n) \sum_i \ell(f, \boldsymbol{x}_i, y_i)$. For each ensemble member, we consider two probability measures: the prior $\pi \in \mathcal{M}(\mathcal{F})$ and the posterior $\hat{\rho} \in \mathcal{M}(\mathcal{F})$. Here, $\mathcal{M}(\mathcal{F})$ denotes the set of all probability measures on $\mathcal{F}$. We encounter cases where we make predictions using the posterior predictive distribution $\mathbf{E}_{f \sim \hat{\rho}}[p(y|\boldsymbol{x}, f)]$. We will use two loss functions, the non-differentiable zero-one loss $\ell_{01}(f, \boldsymbol{x}, y) = \mathbb{I}(\arg\max_j f(\boldsymbol{x})_j \neq y)$, and the negative log-likelihood, which is a commonly used differentiable surrogate $\ell_{\mathrm{nll}}(f, \boldsymbol{x}, y) = -\log(p(y|\boldsymbol{x}, f))$, where we assume that the outputs of $f$ are normalized to form a probability distribution.

Consider the Evidence Lower Bound (ELBO) objective

$$-\mathbf{E}_{f \sim \hat{\rho}} \hat{\mathcal{L}}_{X,Y}^{\ell_{\mathrm{nll}}}(f) - \frac{1}{\lambda n} \mathrm{KL}(\hat{\rho}\|\pi), \tag{1}$$

for some $\lambda > 0$. Catoni [2007] shows that the ELBO is minimized at the probability density given by $\pi(f)e^{-\lambda n \hat{\mathcal{L}}_{X,Y}^{\ell_{\mathrm{nll}}}(f)} / \mathbf{E}_{f \sim \pi}\left[e^{-\lambda n \hat{\mathcal{L}}_{X,Y}^{\ell_{\mathrm{nll}}}(f)}\right]$. We use the Laplace approximation to the posterior in our experiments. This is equivalent to approximating $\hat{\mathcal{L}}_{X,Y}^{\ell_{\mathrm{nll}}}(f)$ using a second-order Taylor expansion around a minimum $\mathbf{w}_{\hat{\rho}}$, such that $\hat{\mathcal{L}}_{X,Y}^{\ell_{\mathrm{nll}}}(f_{\mathbf{w}}) \approx \hat{\mathcal{L}}_{X,Y}^{\ell_{\mathrm{nll}}}(f_{\mathbf{w}_{\hat{\rho}}}) + (\mathbf{w} - \mathbf{w}_{\hat{\rho}})^{\top} \frac{1}{2} \nabla\nabla \hat{\mathcal{L}}_{X,Y}^{\ell_{\mathrm{nll}}}(f_{\mathbf{w}})|_{\mathbf{w}=\mathbf{w}_{\hat{\rho}}} (\mathbf{w} - \mathbf{w}_{\hat{\rho}})$. Assuming a Gaussian prior $\pi = \mathcal{N}(0, \sigma_{\pi}^2 \mathbf{I})$, the Laplace approximation to the posterior $\hat{\rho}$ is again a Gaussian:

$$\hat{\rho} = \mathcal{N}\big(\mathbf{w}_{\hat{\rho}}, \tilde{\mathbf{H}}^{-1}\big),$$

where $\tilde{\mathbf{H}} = \lambda\mathbf{H} + \frac{1}{\sigma_\pi^2}\mathbf{I}$ and $\mathbf{H} = n\nabla\nabla\hat{\mathcal{L}}_{X,Y}^{\ell_{\text{nll}}}(f_\mathbf{w})|_{\mathbf{w}=\mathbf{w}_{\hat{\rho}}}$ is the network Hessian. This Hessian is generally infeasible to compute in practice for modern deep neural networks, such that many approaches employ approximations. Specifically, we will use the Empirical Fisher $\mathbf{F} = \sum_{i=1}^n \nabla_\mathbf{w}\log p(y_i; f, \boldsymbol{x}_i)\nabla_\mathbf{w}\log p(y_i; f, \boldsymbol{x}_i)^\top$, where the labels $y_i$ are the ground-truth labels.

## 3.2 Out-of-sample performance

We now present the following result which links the out-of-sample performance of a linearized Bayesian neural network to increased variance in new data.

**Theorem 1.** *For posterior* $\mathbf{w} \sim \hat{\rho} = \mathcal{N}(\mathbf{w}_{\hat{\rho}}, \tilde{\mathbf{H}}^{-1})$ *and assuming* $\|\mathbf{w}_{\hat{\rho}}\|_2$ *to be bounded, the out-of-sample performance satisfies the following bound*

$$\underbrace{\mathbf{E}_{(y,\boldsymbol{x})\sim\mathcal{D}}\left[-\ln\mathbf{E}_{\mathbf{w}\sim\hat{\rho}}\left[p(y|\boldsymbol{x}, f(\boldsymbol{x}; \mathbf{w}))\right]\right]}_{\text{Bayes Risk}} \lesssim \underbrace{\mathbf{E}_{\mathbf{w}\sim\hat{\rho}}\left[\mathcal{L}_{(y,\boldsymbol{x})\sim\mathcal{D}}^{\ell_{\text{nll}}}(f(\boldsymbol{x}; \mathbf{w}))\right]}_{\text{Gibbs Risk}} \\ - \underbrace{c\mathbf{E}_{(y,\boldsymbol{x})\sim\mathcal{D}}\left[\text{tr}(\mathcal{J}_{\mathbf{w}_{\hat{\rho}}}(\boldsymbol{x})^\top\tilde{\mathbf{H}}^{-1}\mathcal{J}_{\mathbf{w}_{\hat{\rho}}}(\boldsymbol{x}))\right]}_{\text{Variance}}, \tag{2}$$

*for some constant* $c > 0$*, and where the bound* $\lesssim$ *is up to an approximation, and* $\mathcal{J}_{\mathbf{w}_{\hat{\rho}}}(\boldsymbol{x})$ *is the network Jacobian.*

We describe briefly the three terms in the above theorem. The Bayes Risk term is how we evaluate a Bayesian neural network on new data. We take multiple samples from the posterior, compute the average likelihood for each class and then make a prediction. The Gibbs Risk term is the data fitting term we typically use at training time for Bayesian neural networks, *all expectations are outside the log-likelihood*. This term furthermore captures a notion of "flatness", typically seen in Bayesian and PAC-Bayesian objectives [Foret et al., 2020]. The flatter we are the simpler our function is on training data. The Variance term captures something quite different, the variance on new data. The higher the variance, the higher the uncertainty and the tighter the upper bound is on out-of-sample performance. Note also that the Variance term does not require labels.

### Differences from the Masegosa bound

This result is based on a tighter version of Jensen's inequality first introduced in Masegosa [2020]. However crucially, Masegosa [2020] use an objective that requires training all ensemble members jointly with the aim of increasing the ensemble diversity. By contrast, we optimize our objective for each ensemble member individually thus requiring no communication between different ensemble members.

## 3.3 Second order regularization

We first explore how one could avoid dealing with a stochastic neural network at training time. Instead of optimizing a stochastic objective, one can simply fit the mean $f(\boldsymbol{x}; \mathbf{w}_{\hat{\rho}})$ on the labels $y$ for the training set $Z_\text{t}$ and maximize $\text{tr}(\mathcal{J}_{\mathbf{w}_{\hat{\rho}}}(\boldsymbol{x})\tilde{\mathbf{H}}^{-1}\mathcal{J}_{\mathbf{w}_{\hat{\rho}}}(\boldsymbol{x})^\top)$ on unlabeled data $Z_\text{u}$. We then avoid dealing with a stochastic neural network during inference time. Specifically, a general strategy that we will follow is that for each ensemble member $i \in \{1, \cdots, K\}$, we optimize a deterministic objective of the form $\mathcal{L}(\mathbf{w}_{\hat{\rho}}; \cdot) = \hat{\mathcal{L}}_{Z_\text{t}}^{\ell_{\text{nll}}}(f(\boldsymbol{x}; \mathbf{w}_{\hat{\rho}_i})) + R(\mathbf{w}_{\hat{\rho}_i})$, where $R(\mathbf{w}_{\hat{\rho}_i})$ is a regularization term enforcing an appropriate out-of-sample variance. We then construct an ensemble as $\mathcal{E}_K = \{\mathbf{w}_{\hat{\rho}_1}, \cdots, \mathbf{w}_{\hat{\rho}_K}\}$ using the minima obtained by our deterministic objectives. This can be seen as equating each ensemble member with the mean from the approximate Bayesian posterior. We thus use the Bayesian formulation only implicitly, while hoping that the mean captures the desired properties. We finally make predictions using $\frac{1}{K}\sum_{\mathbf{w}_{\hat{\rho}}\in\mathcal{E}_K} p(y|\boldsymbol{x}, f(\boldsymbol{x}; \mathbf{w}_{\hat{\rho}}))$.

We propose two different optimization strategies.

**Strategy 1: Second-order (SO).** For unlabeled samples $Z_\text{u}$, we compute the full Jacobian with respect to the network outputs $\mathcal{J}_{\mathbf{w}_{\hat{\rho}}}(\boldsymbol{x})$. We compute the matrix of variances of all network outputs

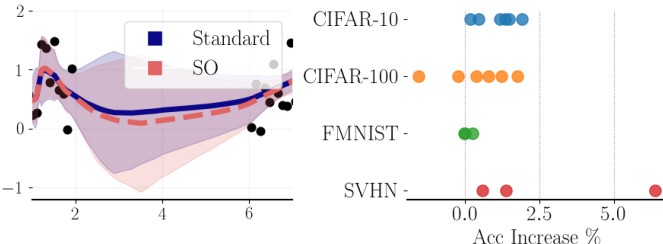
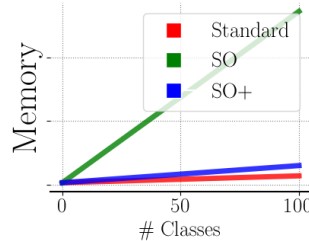

Figure 2: Left: On a toy regression dataset (Section 4.1), and for an ensemble of 20 MLPs with 3 hidden layers, our objective yields increased uncertainty away from the training data, compared with standard deep ensembles. Middle: Improvement of the best of SO/SO+ over Standard ensembles for classification on real datasets (Section 4.2). Our approach yields consistent improvements for all datasets. Right: Additional memory cost of SO and SO+. All methods increase linearly the required memory with the number of classes. However, in the Standard and SO+ methods, we only increase the parameters of the final classification layer. In the SO method, we instead need to compute a full Jacobian per sample instead of a gradient.

$\mathcal{J}_{\mathbf{w}_{\hat{\rho}}}(\boldsymbol{x})\tilde{\mathbf{H}}^{-1}\mathcal{J}_{\mathbf{w}_{\hat{\rho}}}(\boldsymbol{x})^{\top}$ and sum over all diagonal elements $a$. We then optimize

$$\mathcal{L}_{\mathrm{SO}}\left(\mathbf{w}_{\hat{\rho}}; Z_{\mathrm{t}}, Z_{\mathrm{u}}, \beta\right) \triangleq \hat{\mathcal{L}}_{Z_{\mathrm{t}}}^{\ell_{\mathrm{nll}}}(f(\boldsymbol{x}; \mathbf{w}_{\hat{\rho}})) - \beta \sum_{\boldsymbol{x}, y \in Z_{\mathrm{u}}} \sum_{a} \left[\mathcal{J}_{\mathbf{w}_{\hat{\rho}}}(\boldsymbol{x})^{\top}\tilde{\mathbf{H}}^{-1}\mathcal{J}_{\mathbf{w}_{\hat{\rho}}}(\boldsymbol{x})\right]_{a}, \quad (3)$$

where $\beta$ is the regularization strength. See Algorithm 1 in Appendix A. There are several costs associated with this objective as it requires the computation of the *full network Jacobian per unlabeled sample*: 1) our memory costs increase by a factor of $|\mathcal{Y}|$, where $|\mathcal{Y}|$ is the number of classes, see right panel of Figure 2; 2) there are increased computational costs associated with computing Jacobians *and* gradients per sample. Most deep learning libraries are optimized to compute gradient information, on the batch and not the individual sample level.

We manage to alleviate costs 1) described above through the following batching strategy.

**Strategy 2: Randomized second-order (SO+).** For each unlabeled sample $(\boldsymbol{x}, \cdot) \in Z_{\mathrm{u}}$, we compute the gradient with respect to some random network output $f_{a_{\boldsymbol{x}}}(\boldsymbol{x}; \mathbf{w})$ where $a_{\boldsymbol{x}} \sim \mathrm{Cat}(|\mathcal{Y}|, 1/|\mathcal{Y}|)$. We then optimize

$$\mathcal{L}_{\mathrm{SO+}}\left(\mathbf{w}_{\hat{\rho}}; Z_{\mathrm{t}}, Z_{\mathrm{u}}, \beta\right) \triangleq \hat{\mathcal{L}}_{Z_{\mathrm{t}}}^{\ell_{\mathrm{nll}}}(f(\boldsymbol{x}; \mathbf{w}_{\hat{\rho}})) - \beta \sum_{\boldsymbol{x}, y \in Z_{\mathrm{u}}} \left[\mathcal{J}_{\mathbf{w}_{\hat{\rho}}}(\boldsymbol{x})^{\top}\tilde{\mathbf{H}}^{-1}\mathcal{J}_{\mathbf{w}_{\hat{\rho}}}(\boldsymbol{x})\right]_{a_{\boldsymbol{x}}}. \quad (4)$$

By sampling some random class $a_{\boldsymbol{x}}$ for which to compute the gradient per sample, we avoid incurring the $\times|\mathcal{Y}|$ cost associated with computing the full Jacobian. We decrease the memory cost correspondingly. The drawback is that we further approximate our estimate of the variance, thus possibly converging to worse solutions. See Algorithm 2 in Appendix A.

### 3.4 Heuristic gradient and batching

The regularization terms considered in the previous section require computing a gradient with respect to the parameters when learning the model. However, such gradients involve differentiating w.r.t. to the Hessian, which requires third-order derivatives and is therefore infeasible in practice for large models. One could replace the Hessian with the Fisher matrix which would result in second-order derivatives and is still expensive. Instead, we propose to use cheaper *regularization vector fields* when updating the model's parameters. We construct these vector fields by using an ansatz inspired by the expression of the gradient of the regularization terms. In the case of the SO term, the diagonal Empirical Fisher approximation to the Hessian, the vector field takes the following form:

$$\Gamma(\mathbf{w})_j = -2\lambda \sum_i A_i(Z_{\mathrm{t}})^{-2} \left( \sum_{\boldsymbol{x}_{\mathrm{t}}, y_{\mathrm{t}} \in Z_{\mathrm{t}}} U(\boldsymbol{x}_{\mathrm{t}}, \mathbf{w})_{i,j} \frac{\partial \log p(y_{\mathrm{t}}; f)}{\partial \mathbf{w}_i} \right) \sum_{\boldsymbol{x}_{\mathrm{u}} \in Z_{\mathrm{u}}} \sum_a \left( \frac{\partial f_a(\boldsymbol{x}_{\mathrm{u}}; \mathbf{w})}{\partial \mathbf{w}_i} \right)^2$$

$$+ 2 \sum_i A_i(Z_{\mathrm{t}})^{-1} \sum_{\boldsymbol{x}_{\mathrm{u}} \in Z_{\mathrm{u}}} \sum_a \left( V_a(\boldsymbol{x}_{\mathrm{u}}, \mathbf{w})_{i,j} \frac{\partial f_a(\boldsymbol{x}_{\mathrm{u}}; \mathbf{w})}{\partial \mathbf{w}_i} \right),$$

$$(5)$$

where $A_i(Z_t) = \frac{1}{\sigma_\pi^2} + \lambda \sum_{\boldsymbol{x}_t, y_t \in Z_t} \frac{\partial \log p(y_t|f)}{\partial \mathbf{w}_i}$, and $U$ and $V_a$ are matrices to be chosen. For specific choices of $U$ and $V_a$, the vector field $\Gamma$ recovers the gradient of the SO term introduced in the previous section. Instead of computing these exact derivatives, we choose $U$ and $V_a$ to be diagonal and to contain only first-order derivatives. Specifically, they take the following form:

$$U(\boldsymbol{x}_t; \mathbf{w})_{i,i} = \phi \left[ \frac{\partial \log p(y_t; f)}{\partial \mathbf{w}_i} \right]^2, \qquad V_a(\boldsymbol{x}_u; \mathbf{w})_{i,i} = \chi \left[ \frac{\partial f_a(\boldsymbol{x}_u, \mathbf{w})}{\partial \mathbf{w}_i} \right]^2. \qquad (6)$$

and we restrict $\sum_i$ to $\sum_{i=j}$. We discuss this choice, as well as the effect of the parameters $\phi, \chi$ and provide more details on the derivation in Appendix C. Unless stated otherwise we will use $\phi = 1, \chi = 1$. For this choice our regulariser has a simple and intuitive effect which we plot in Figure 1. Consider a single training $\boldsymbol{x}_t$ and unlabeled sample $\boldsymbol{x}_u$, and a single weight $w$: i) Non-zero gradient steps, ie $\frac{\partial}{\partial w}[R(\mathbf{w})] \neq 0$, are taken only when $\frac{\partial p(y; \boldsymbol{x}_t; \mathbf{w})}{\partial w} \approx 0$ and $\frac{\partial f_a(\boldsymbol{x}_u; \mathbf{w})}{\partial w} \neq 0$, i.e. our regulariser is activated *as long as this does not interfere with fitting the labels on the training set.* ii) $-\frac{\partial}{\partial w}[R(\mathbf{w})]$ and $\frac{\partial f_a(\boldsymbol{x}_u; \mathbf{w})}{\partial w}$ have the same sign. Intuitively, this means that our regulariser (when activated) *encourages that we fit all classes (SO) or a random class a (SO+) for each unlabeled sample* (we move in the direction that maximizes the corresponding logit).

We finally see that it is easy to parallelize the above by substituting with $B_t$ a minibatch of training data, and $B_u$ a minibatch of unlabeled data. We need to compute $\frac{\partial \log p(y_t; f)}{\partial \mathbf{w}_i}$ per training sample $(\boldsymbol{x}_t, y_t)$. We also need to compute $\frac{\partial f_a(\boldsymbol{x}_u; \mathbf{w})}{\partial \mathbf{w}_i}$ per unlabeled sample $(\boldsymbol{x}_u, \cdot)$ and per output $a$. Since Equation (22) is *per weight $j$*, the full gradient can be computed with simple multiplications and additions, and is vectorisable. Finally on the computational side, per-sample gradient estimation and Jacobian estimation is possible in both PyTorch and JAX.

## 4 Experiments

In this section, we present a number of experiments that investigate whether the proposed regularizers improve deep ensembles.

### 4.1 Toy regression example

We created a toy 1d regression dataset with input $x \in [0, 10]$ and output distributed as $y \sim \mathcal{N}(\sin(x), 1)$, a training set of size $|Z_t| = 80$ and an unlabeled set of size $|Z_u| = 100$. For the regression architecture, we used an MLP with 3 hidden layers each with 100 neurons. We fit the training set with an ensemble of 20 networks using the Standard Ensemble approach of Lakshminarayanan et al. [2017] which enforces diversity with different random seeds and with our SO objective. We then plot the resulting predictive mean as well as a $\pm 2\sigma$ confidence interval in the left panel of Figure 2. We see that both the Standard and the SO ensembles fit the training data equally well. However, the SO ensemble has a significantly higher uncertainty away from training data.

### 4.2 Classification on real datasets

We train ensembles on 4 different datasets, CIFAR-10, CIFAR-100, Fashion MNIST and SVHN. We used 3 different architectures, the WideResNet22 architecture, as well as an MLP, and the LeNet architecture. For the MLP we used 3 hidden layers with 784, 500 and 300 neurons. The training dataset size was 40000 samples. For all architectures, we used 10000 samples for testing and 1000 samples for validation. The remaining samples were used as a pool for unlabeled data. This dataset split was first proposed in Alayrac et al. [2019]. The dataset split is especially difficult for CIFAR-100 as very few labeled samples would be available per class for the 1000 training sample case, and in the 100 sample case, some classes would have no training samples. Hyperparameter optimization was done using the Optuna Bayesian Optimizer. We compare with Standard ensembles [Lakshminarayanan et al., 2017] (trained with independent seeds), Masegosa ensembles [Masegosa, 2020] which optimize diversity in function space, DICE ensembles [Ramé and Cord, 2021] which is currently the SOTA method in diverse ensemble training, and Agree to Disagree (A2D) ensembles [Matteo et al., 2023]. Unless stated otherwise, the ensemble size for all methods is 10 ensemble members, which is one of the most common sizes used in practice. We also experiment with the

Table 1: **WideResNet22 ablations.** SO/SO+ often improves over standard ensembles. When it doesn't improve up standard ensembles SO has typically almost the same test metrics as standard ensembles. We highlight the cases where SO strictly improves upon standard ensembles. Hyperparameter optimization was done using random search.

| Dataset / Aug | Method | Acc ↑ | ECE ↓ | TACE ↓ | Brier ↓ | NLL ↓ |
|---|---|---|---|---|---|---|
| CIFAR-10 | Standard40K | 84.52 | 0.04 | 0.01 | 0.86 | 0.73 |
| / no augmentation | SO40K@1K | 85.29 | 0.04 | 0.01 | 0.86 | 0.57 |
| | SO40K@5K | 84.42 | 0.04 | 0.011 | 0.84 | 0.59 |
| | SO+40K@1K | 85.90 | 0.04 | 0.011 | 0.87 | 0.59 |
| | A2D40K | 85.34 | 0.04 | 0.013 | 0.86 | 0.72 |
| | Masegosa40K | 83.07 | 0.27 | 0.05 | 0.44 | 0.78 |
| CIFAR-100 | Standard40K | 51.30 | 0.14 | 0.004 | 0.57 | 2.66 |
| / no augmentation | SO40K@1K | 47.85 | 0.08 | 0.003 | 0.43 | 2.18 |
| | SO40K@5K | 49.66 | 0.03 | 0.002 | 0.39 | 1.93 |
| | SO+40K@1K | 54.14 | 0.17 | 0.004 | 0.64 | 2.95 |
| | A2D40K | 51.64 | 0.16 | 0.004 | 0.60 | 2.96 |
| | Masegosa40K | 49.67 | 0.11 | 0.003 | 0.23 | 1.96 |
| FMNIST | Standard40K | 92.66 | 0.03 | 0.006 | 0.94 | 0.31 |
| / no augmentation | SO40K@1K | 92.49 | 0.018 | 0.004 | 0.91 | 0.22 |
| | SO40K@5K | 92.67 | 0.03 | 0.006 | 0.95 | 0.38 |
| | A2D40K | 92.37 | 0.03 | 0.006 | 0.60 | 0.45 |
| | Masegosa40K | 93.02 | 0.31 | 0.059 | 0.47 | 0.53 |
| SVHN | Standard40K | 95.72 | 0.016 | 0.003 | 0.95 | 0.25 |
| / no augmentation | SO40K@1K | 95.68 | 0.014 | 0.003 | 0.95 | 0.24 |
| | SO40K@5K | 95.60 | 0.01 | 0.003 | 0.95 | 0.26 |
| | A2D40K | 93.17 | 0.13 | 0.024 | 0.68 | 0.34 |
| | Masegosa40K | 95.63 | 0.33 | 0.065 | 0.49 | 0.55 |
| CIFAR-10 | Standard40K | 91.67 | 0.031 | 0.006 | 0.92 | 0.39 |
| / flip + crops | SO+40K@1K | 91.24 | 0.033 | 0.007 | 0.92 | 0.39 |
| | $\nu$-ens40K@1K | 91.87 | 0.033 | 0.007 | 0.93 | 0.40 |
| CIFAR-100 | Standard40K | 66.38 | 0.12 | 0.003 | 0.74 | 2.20 |
| / flip + crops | SO+40K@1K | 66.33 | 0.13 | 0.003 | 0.75 | 2.30 |
| | $\nu$-ens40K@1K | 65.74 | 0.12 | 0.003 | 0.73 | 2.11 |

standard setup of augmentations for CIFAR-10 and CIFAR-100. Specifically, we augment the training set with random flips and crops.

We expect our approach to improve testing accuracy as well as the calibration of predictions. We thus evaluate the final ensembles on test Accuracy, the Expected Calibrations Error (ECE), the Thresholded Calibration Error (TACE), the Brier score, as well as the test Negative Log-Likelihood (NLL). It is folk wisdom that calibration can be traded for accuracy when using the ECE and the TACE. As a consequence, we also find the Pareto optimal Stiglitz [1981] ensemble in terms of both accuracy and calibration. Specifically we find the ensemble that minimizes $(1 - \text{Accuracy})^2 + (\text{TACE})^2$. When the TACE is not available, which can happen due to binning, we use the ECE instead. We include a short description of Pareto optimality in Appendix D.

For SO we use all the remaining samples as a pool for training time. Sweeping through the entire unlabeled set would be unfeasible. Instead, we simply sample at each iteration an unlabeled batch of the same size as the training batch, and take a gradient step. Since SO+ is much more memory efficient than SO, we select an unlabeled set and we sweep through all of it at each epoch. We present the results in Tables 1 and 2. See also Table 1 in Appendix F.

**SO/SO+ improve both accuracy and calibration.** We often see significant improvements with our approach over both Standard ensembles and the other diversity approaches in all setups (see the middle panel of Figure 2). We gain up to $6.38\%$ in test Accuracy depending on the architecture and

Table 2: **LeNet - MLP ablations.** SO/SO+ often improves over standard ensembles. When it doesn't improve up standard ensembles SO has typically almost the same test metrics as standard ensembles. Note that accuracy can be traded off for calibration, as such we consider that a method is better than the other only if it improves upon both metrics. We highlight the cases where SO strictly improves upon standard ensembles. Hyperparameter optimization was done using random search.

| Dataset / Model | Method | Acc ↑ | ECE ↓ | TACE ↓ | Brier ↓ | NLL ↓ |
|---|---|---|---|---|---|---|
| CIFAR-10 | Standard40K | 71.25 | 0.10 | 0.02 | 0.77 | 1.41 |
| / LeNet | SO40K@1K | 71.36 | 0.11 | 0.024 | 0.79 | 2.25 |
| / no augmentation | SO40K@5K | 71.63 | 0.10 | 0.023 | 0.77 | 1.41 |
| | SO+40K@1K | 70.60 | 0.11 | 0.025 | 0.79 | 2.13 |
| | A2D40K | 71.34 | 0.06 | 0.016 | 0.70 | 0.94 |
| | Masegosa40K | 68.97 | 0.17 | 0.03 | 0.39 | 1.00 |
| CIFAR-10 | Standard40K | 55.84 | 0.21 | 0.04 | 0.71 | 2.55 |
| / MLP | SO40K@1K | 56.00 | 0.23 | 0.041 | 0.74 | 3.09 |
| / no augmentation | SO40K@5K | 55.61 | 0.22 | 0.040 | 0.72 | 2.71 |
| | SO+40K@1K | 56.23 | 0.22 | 0.04 | 0.73 | 2.94 |
| | A2D40K | 54.14 | 0.20 | 0.041 | 0.68 | 2.74 |
| | Masegosa40K | 53.04 | 0.04 | 0.019 | 0.40 | 1.35 |
| CIFAR-100 | Standard40K | 36.38 | 0.23 | 0.005 | 0.49 | 4.44 |
| / LeNet | SO40K@1K | 37.28 | 0.11 | 0.003 | 0.35 | 2.81 |
| / no augmentation | SO40K@5K | 36.74 | 0.12 | 0.004 | 0.36 | 2.86 |
| | SO+40K@1K | 37.46 | 0.11 | 0.003 | 0.35 | 2.8 |
| | A2D40K | 35.92 | 0.12 | 0.0041 | 0.35 | 2.96 |
| | Masegosa40K | 34.70 | 0.05 | 0.003 | 0.23 | 2.75 |
| CIFAR-100 | Standard40K | 27.43 | 0.23 | 0.006 | 0.38 | 4.33 |
| / MLP | SO40K@1K | 28.25 | 0.18 | 0.005 | 0.33 | 3.79 |
| / no augmentation | SO40K@5K | 27.68 | 0.19 | 0.005 | 0.33 | 3.78 |
| | SO+40K@1K | 27.75 | 0.23 | 0.005 | 0.38 | 4.25 |
| | A2D40K | 28.02 | 0.06 | 0.0032 | 0.21 | 3.11 |
| | Masegosa40K | 26.66 | 0.12 | 0.004 | 0.24 | 3.44 |
| FMNIST | Standard40K | 92.50 | 0.039 | 0.006 | 0.039 | 0.49 |
| / LeNet | SO40K@1K | 92.19 | 0.022 | 0.005 | 0.91 | 0.24 |
| / no augmentation | SO40K@5K | 92.43 | 0.03 | 0.006 | 0.94 | 0.31 |
| | A2D40K | 91.95 | 0.05 | 0.009 | 0.91 | 0.61 |
| | Masegosa40K | 91.69 | 0.29 | 0.055 | 0.49 | 0.53 |
| FMNIST | Standard40K | 89.35 | 0.029 | 0.005 | 0.89 | - |
| / MLP | SO40K@1K | 89.31 | 0.021 | 0.004 | 0.88 | - |
| / no augmentation | SO40K@5K | 89.48 | 0.05 | 0.009 | 0.9355 | 0.52 |
| | A2D40K | 88.44 | 0.02 | 0.006 | 0.87 | 0.33 |
| | Masegosa40K | 89.50 | 0.27 | 0.056 | 0.50 | 0.58 |
| SVHN | Standard40K | 87.98 | 0.04 | 0.01 | 0.90 | - |
| / LeNet | SO40K@1K | 87.90 | 0.04 | 0.01 | 0.90 | - |
| / no augmentation | SO40K@5K | 88.10 | 0.045 | 0.009 | 0.92 | - |
| | A2D40K | 86.11 | 0.03 | 0.011 | 0.85 | 0.98 |
| | Masegosa40K | 86.21 | 0.26 | 0.047 | 0.45 | 0.70 |
| SVHN | Standard40K | 82.13 | 0.01 | 0.008 | 0.76 | 0.66 |
| / MLP | SO40K@1K | 82.52 | 0.04 | 0.01 | 0.82 | 0.77 |
| / no augmentation | SO40K@5K | 82.10 | 0.053 | 0.012 | 0.83 | 0.89 |
| | A2D40K | 83.10 | 0.03 | 0.008 | 0.81 | 0.68 |
| | Masegosa40K | 78.66 | 0.23 | 0.041 | 0.41 | 0.91 |

dataset. We are Pareto optimal in 12 out of 18 cases and are optimal in terms of test accuracy in 12 out of 18 cases. The rest of the "wins" are split among the other approaches.

**SO/SO+ is much easier to optimize than alternatives.** One particular advantage is that when we do not improve upon other methods, we typically do not hurt performance significantly. This is in stark contrast with DICE. We had difficulties getting DICE to converge on cases other than MLP/LeNet for CIFAR-10. After correspondence with the authors, we believe that it is difficult to find the hyperparameters for which DICE works. It is folk wisdom that adversarial objectives are in general difficult to tune. We also observe that Masegosa and A2D ensembles typically underfit. In some cases though they do improve upon Standard Ensembles. We emphasize that in the A2D case, we replicated the results on the M/F-Dominoes and M/C-Dominoes datasets from the original paper. This means that the underfitting we report is not due to our implementation, but inherent in the algorithm.

**SO/SO+ does not require communication between ensemble members.** We trained each ensemble member independently and only evaluated the complete ensemble at the end of training for validation and testing.

**SO+ can scale to moderate dataset sizes.** We used SO+ to scale to 20K unlabeled samples per epoch for the MLP architecture, 10K unlabeled samples for the LeNet architecture, and 1-5K samples for the WideResNet22 architecture. We note that in principle one can scale to larger sets, as each ensemble member is trained independently and with stochastic batches.

**SO/SO+ improve ensembles even when using data augmentation.** Regularization gains in the low data regime often vanish or are greatly reduced when we apply data augmentation. In Tables 1 and 2, we see that augmentations improve the standard deep ensembles and that SO/SO+ provide further improvements.

## 5 Limitations

We introduced a number of approximations so as to obtain a tractable regulariser. These could mean that our results are suboptimal compared to what could be achieved with less restrictive modeling. We managed to scale our approach to moderate dataset sizes. However, our approach incurs an increase in computational time. Also the number of training *classes* increases linearly our memory cost.

## 6 Conclusion

We introduced SO and SO+, two novel objectives that improve deep ensembles by leveraging unlabeled data. SO and SO+ do not require communication between the different ensemble members during training, beat consistently other approaches both in terms of accuracy and calibration, and avoid the stability issues of other objectives. A key question is if we can obtain a more principled regulariser and whether it can result in improved results. In future work, we will also try to make our approach even more scalable.

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

# 7 Pseudocode

We provide the pseudocode algorithms for the SO and SO+ strategies described in the main text.

---

**Algorithm 1** Pseudocode for the SO algorithm

---

**Input:** Temperature $\lambda$, regularisation coefficient $\beta$, prior variance $\sigma_\pi^2$, training data $Z_\mathrm{t}$, unlabeled data $Z_\mathrm{u}$, number of ensemble members $K$

**Output:** Ensemble $\mathcal{E}_K = \{\mathbf{w}_{\hat{\rho}_1}, \ldots, \mathbf{w}_{\hat{\rho}_K}\}$

   $\mathcal{E}_K \leftarrow \{\}$

   **for** $i$ in $\{1, \ldots, K\}$ **do**

      $\mathbf{w}_{\hat{\rho}} \leftarrow$ Random Initialization

      **while** not converged **do**

         Draw labeled samples $B_\mathrm{t}$ and unlabeled samples $B_\mathrm{u}$

         Compute Empirical Fisher matrix $\mathbf{F} \leftarrow \sum_{(\boldsymbol{x}_t, y_t) \in B_\mathrm{t}} \nabla_{\mathbf{w}_{\hat{\rho}}} \log p(y_t; f, \boldsymbol{x}_t) \nabla_{\mathbf{w}_{\hat{\rho}}} \log p(y_t; f, \boldsymbol{x}_t)^\top$

         Compute the diagonal approximation to the Hessian $\tilde{\mathbf{H}} \leftarrow \lambda \mathrm{diag}(\mathbf{F}) + \frac{1}{\sigma_\pi^2} \mathbf{I}$

         Compute the Empirical Risk $\hat{\mathcal{L}}_{B_\mathrm{t}}^{\ell_\mathrm{nll}}(f(\boldsymbol{x}; \mathbf{w}_{\hat{\rho}}))$

         Compute the heuristic gradient

$$\Gamma_{\mathrm{SO}}(\mathbf{w})_j = -2\lambda A_j(Z_\mathrm{t})^{-2} \left( \sum_{\boldsymbol{x}_t, y_t \in Z_\mathrm{t}} \phi \left[ \frac{\partial \log p(y_t; f)}{\partial \mathbf{w}_j} \right]^3 \right) \sum_{\boldsymbol{x}_\mathrm{u} \in Z_\mathrm{u}} \sum_a \left( \frac{\partial f_a(\boldsymbol{x}_\mathrm{u}; \mathbf{w})}{\partial \mathbf{w}_j} \right)^2$$

$$+ 2A_j(Z_\mathrm{t})^{-1} \sum_{\boldsymbol{x}_\mathrm{u} \in Z_\mathrm{u}} \sum_a \left( \chi \left[ \frac{\partial f_a(\boldsymbol{x}_\mathrm{u}, \mathbf{w})}{\partial \mathbf{w}_j} \right]^3 \right),$$

         where $A_j(Z_\mathrm{t}) = \frac{1}{\sigma_\pi^2} + \lambda \sum_{\boldsymbol{x}_t, y_t \in Z_\mathrm{t}} \left( \frac{\partial \log p(y_t; f)}{\partial \mathbf{w}_j} \right)^2$ and $\phi = \chi = 1$.

         Update with gradient $g = \nabla_{\mathbf{w}_{\hat{\rho}}} \hat{\mathcal{L}}_{B_\mathrm{t}}^{\ell_\mathrm{nll}}(f(\boldsymbol{x}; \mathbf{w}_{\hat{\rho}})) - \beta \Gamma_{\mathrm{SO}}(\mathbf{w}_{\hat{\rho}})$

      **end while**

      $\mathcal{E}_K \leftarrow \mathcal{E}_K \cup \{\mathbf{w}_{\hat{\rho}}\}$

   **end for**

---

**Algorithm 2** Pseudocode for the SO+ algorithm

---

**Input:** Temperature $\lambda$, regularisation coefficient $\beta$, prior variance $\sigma_\pi^2$, training data $Z_\mathrm{t}$, unlabeled data $Z_\mathrm{u}$, number of ensemble members $K$
**Output:** Ensemble $\mathcal{E}_K = \{\mathbf{w}_{\hat{\rho}_1}, \ldots, \mathbf{w}_{\hat{\rho}_K}\}$

  $\mathcal{E}_K \leftarrow \{\}$
  **for** $i$ in $\{1, \ldots, K\}$ **do**
      $\mathbf{w}_{\hat{\rho}} \leftarrow$ Random Initialization
      **while** not converged **do**
          Draw labeled samples $B_\mathrm{t}$ and unlabeled samples $B_\mathrm{u}$
          Compute Empirical Fisher matrix $\mathbf{F} \leftarrow \sum_{(\boldsymbol{x}_t, y_t) \in B_\mathrm{t}} \nabla_{\mathbf{w}_{\hat{\rho}}} \log p(y_t; f, \boldsymbol{x}_t) \nabla_{\mathbf{w}_{\hat{\rho}}} \log p(y_t; f, \boldsymbol{x}_t)^\top$
          Compute the diagonal approximation to the Hessian $\tilde{\mathbf{H}} \leftarrow \lambda \mathrm{diag}(\mathbf{F}) + \frac{1}{\sigma_\pi^2} \mathbf{I}$
          $\forall (\boldsymbol{x}_u, \cdot) \in B_\mathrm{u}$ sample $a_{\boldsymbol{x}} \sim \mathrm{Cat}(|\mathcal{Y}|, 1/|\mathcal{Y}|)$
          Compute the Empirical Risk $\hat{\mathcal{L}}_{B_\mathrm{t}}^{\ell_{\mathrm{nll}}}(f(\boldsymbol{x}; \mathbf{w}_{\hat{\rho}}))$
          Compute the heuristic gradient

$$
\Gamma_{\mathrm{SO+}}(\mathbf{w})_j = -2\lambda A_j(Z_\mathrm{t})^{-2} \left( \sum_{\boldsymbol{x}_t, y_t \in Z_\mathrm{t}} \phi \left[ \frac{\partial \log p(y_t; f)}{\partial \mathbf{w}_j} \right]^3 \right) \sum_{\boldsymbol{x}_u \in Z_\mathrm{u}} \left( \frac{\partial f_{a_{\boldsymbol{x}}}(\boldsymbol{x}_\mathrm{u}; \mathbf{w})}{\partial \mathbf{w}_j} \right)^2
$$
$$
+ 2 A_j(Z_\mathrm{t})^{-1} \sum_{\boldsymbol{x}_u \in Z_\mathrm{u}} \left( \chi \left[ \frac{\partial f_{a_{\boldsymbol{x}}}(\boldsymbol{x}_\mathrm{u}, \mathbf{w})}{\partial \mathbf{w}_j} \right]^3 \right),
$$

          where $A_j(Z_\mathrm{t}) = \frac{1}{\sigma_\pi^2} + \lambda \sum_{\boldsymbol{x}_t, y_t \in Z_\mathrm{t}} \left( \frac{\partial \log p(y_t; f)}{\partial \mathbf{w}_j} \right)^2$ and $\phi = \chi = 1$.
          Update with gradient $g = \nabla_{\mathbf{w}_{\hat{\rho}}} \hat{\mathcal{L}}_{B_\mathrm{t}}^{\ell_{\mathrm{nll}}}(f(\boldsymbol{x}; \mathbf{w}_{\hat{\rho}})) - \beta \Gamma_{\mathrm{SO+}}(\mathbf{w}_{\hat{\rho}})$
      **end while**
      $\mathcal{E}_K \leftarrow \mathcal{E}_K \cup \{\mathbf{w}_{\hat{\rho}}\}$
  **end for**

---

# 8 Motivating the optimization objective through a PAC-Bayes bound

We now prove Theorem 1 which indicates an appropriate way to choose $\mathbf{w}_{\hat{\rho}}$ such as to ensure good out-of-sample performance.

**Theorem 2.** *For posterior $\mathbf{w} \sim \hat{\rho} = \mathcal{N}\left(\mathbf{w}_{\hat{\rho}}, \tilde{\mathbf{H}}^{-1}\right)$ and assuming $\|\mathbf{w}_{\hat{\rho}}\|_2$ to be bounded, the out-of-sample performance satisfies the following bound*

$$
\underbrace{\mathbf{E}_{(y,\boldsymbol{x})\sim\mathcal{D}}\left[-\ln \mathbf{E}_{\mathbf{w}\sim\hat{\rho}}\left[p(y|\boldsymbol{x}, f(\boldsymbol{x};\mathbf{w}))\right]\right]}_{\text{Bayes Risk}} \lesssim \underbrace{\mathbf{E}_{\mathbf{w}\sim\hat{\rho}}\left[\mathcal{L}_{(y,\boldsymbol{x})\sim\mathcal{D}}^{\ell_{\text{nll}}}(f(\boldsymbol{x};\mathbf{w}))\right]}_{\text{Gibbs Risk}}
$$
$$
\underbrace{- c\mathbf{E}_{(y,\boldsymbol{x})\sim\mathcal{D}}\left[\text{tr}(\mathcal{J}_{\mathbf{w}_{\hat{\rho}}}(\boldsymbol{x})^\top \tilde{\mathbf{H}}^{-1} \mathcal{J}_{\mathbf{w}_{\hat{\rho}}}(\boldsymbol{x}))\right]}_{\text{Variance}},
\tag{7}
$$

*for some constant $c > 0$, and where the bound $\lesssim$ is up to an approximation, and $\mathcal{J}_{\mathbf{w}_{\hat{\rho}}}(\boldsymbol{x})$ is the network Jacobian.*

*Proof.* The proof is based on the following theorem that links generalization with a variance of the posterior predictive.

**Theorem 3.** *Masegosa [2020] Any distribution $\hat{\rho}$ in the space of distributions $\mathcal{M}$ satisfies that,*

$$
\mathbf{E}_{(y,\boldsymbol{x})\sim\mathcal{D}}\left[-\ln \mathbf{E}_{\mathbf{w}\sim\hat{\rho}}\left[p(y|\boldsymbol{x}, f(\boldsymbol{x};\mathbf{w}))\right]\right] \leq \mathbf{E}_{\mathbf{w}\sim\hat{\rho}}\left[\mathcal{L}_{(y,\boldsymbol{x})\sim\mathcal{D}}^{\ell_{\text{nll}}}(f(\boldsymbol{x};\mathbf{w}))\right] - \mathbf{V}(\hat{\rho})
\tag{8}
$$

*where $\mathbf{V}(\hat{\rho})$ is a variance term defined as*

$$
\mathbf{V}(\hat{\rho}) = \mathbf{E}_{(y,\boldsymbol{x})\sim\mathcal{D}}\left[\frac{1}{2\max_{\mathbf{w}} p(y|\boldsymbol{x};\mathbf{w})}\mathbf{E}_{\mathbf{w}\sim\hat{\rho}}\left[(p(y|\boldsymbol{x},\mathbf{w}) - \mathbf{E}_{\mathbf{w}\sim\hat{\rho}}(p(y|\boldsymbol{x},\mathbf{w})))^2\right]\right].
\tag{9}
$$

Let us assume as in Masegosa [2020] that the model likelihood is bounded:

**Assumption 1.** *Masegosa [2020] There exists a constant $C < \infty$ such that $\forall \boldsymbol{x} \in \mathcal{X}$, $\max_{y,\mathbf{w}} p(y|\boldsymbol{x};\mathbf{w}) \leq C$.*

Note that this assumption holds for the classification setting with $C = 1$.

The first term in the RHS of (8) is the common data fitting term when optimizing a supervised classification objective. As such we will focus on the second "variance" term $\mathbf{V}(\hat{\rho})$.

We use a linearization of the neural network outputs, such that

$$
f_{\text{lin}}(\boldsymbol{x};\mathbf{w}) = f(\boldsymbol{x};\mathbf{w}_{\hat{\rho}}) + \mathcal{J}_{\mathbf{w}_{\hat{\rho}}}(\boldsymbol{x})^\top (\mathbf{w} - \mathbf{w}_{\hat{\rho}}).
\tag{10}
$$

Then for posterior $\mathbf{w} \sim \hat{\rho} = \mathcal{N}\left(\mathbf{w}_{\hat{\rho}}, \tilde{\mathbf{H}}^{-1}\right)$ the outputs of the linearized network have the following distribution

$$
f_{\text{lin}}(\boldsymbol{x}) \sim p(f_{\text{lin}}(\boldsymbol{x})|\boldsymbol{x}, X, Y) = \mathcal{N}(f(\boldsymbol{x};\mathbf{w}_{\hat{\rho}}), \mathcal{J}_{\mathbf{w}_{\hat{\rho}}}(\boldsymbol{x})^\top \tilde{\mathbf{H}}^{-1} \mathcal{J}_{\mathbf{w}_{\hat{\rho}}}(\boldsymbol{x}))
$$

We use the cross-entropy loss. We take a first order approximation to the loss. Then for the $f_{\text{lin}}$ predictor and the variance term $\mathbf{V}(\hat{\rho})$ we have that

$$
\mathbf{V}(\hat{\rho}) \geq \frac{1}{2}\mathbf{E}_{(y,\boldsymbol{x})\sim\mathcal{D}}\left[\mathbf{E}_{\mathbf{w}\sim\hat{\rho}}\left[(p(y|\boldsymbol{x},\mathbf{w}) - \mathbf{E}_{\mathbf{w}\sim\hat{\rho}}(p(y|\boldsymbol{x},\mathbf{w})))^2\right]\right]
$$
$$
\approx \frac{1}{2}\mathbf{E}_{(y,\boldsymbol{x})\sim\mathcal{D}}\left[\mathbf{E}_{f_{\text{lin}}(\boldsymbol{x})\sim p(f_{\text{lin}}(\boldsymbol{x})|\boldsymbol{x},X,Y)}\left[(p(y|f_{\text{lin}}(\boldsymbol{x})) - \mathbf{E}_{f_{\text{lin}}(\boldsymbol{x})\sim p(f_{\text{lin}}(\boldsymbol{x})|\boldsymbol{x},X,Y)}\left[p(y|f_{\text{lin}}(\boldsymbol{x}))\right])^2\right]\right]
$$
$$
\approx \frac{1}{2}\mathbf{E}_{(y,\boldsymbol{x})\sim\mathcal{D}}\left[\mathcal{J}_f(\boldsymbol{x})^\top \left[\mathcal{J}_{\mathbf{w}_{\hat{\rho}}}(\boldsymbol{x})^\top \tilde{\mathbf{H}}^{-1} \mathcal{J}_{\mathbf{w}_{\hat{\rho}}}(\boldsymbol{x})\right] \mathcal{J}_f(\boldsymbol{x})\right]
\tag{11}
$$

where $\mathcal{J}_f(\boldsymbol{x})_i = \left.\frac{\partial p(y;f)}{\partial f_i}\right|_{f=f(\boldsymbol{x};\mathbf{w}_{\hat{\rho}})}$. The first inequality follows from the inequality $p(y;\boldsymbol{x},\mathbf{w}) \leq 1$.
The second line is obtained using the first-order approximation of $f(x,w)$ ($f(\boldsymbol{x};\mathbf{w}) \approx f_{lin}(\boldsymbol{x};\mathbf{w})$.

Such an approximation is controlled by the variance of $\mathbf{w}$ which is of order $1/N$. Finally, the last line results from the application of the delta method which also holds with an error of order $1/N$.

We now deal with this Jacobian term which is the only one dependent on the labels $y$.

$$
\begin{aligned}
\mathbf{V}(\hat{\rho}) &\gtrapprox \frac{1}{2}\mathbf{E}_{(y,\boldsymbol{x})\sim\mathcal{D}}\left[\mathcal{J}_f(\boldsymbol{x})^\top\left[\mathcal{J}_{\mathbf{w}_{\hat{\rho}}}(\boldsymbol{x})^\top\tilde{\mathbf{H}}^{-1}\mathcal{J}_{\mathbf{w}_{\hat{\rho}}}(\boldsymbol{x})\right]\mathcal{J}_f(\boldsymbol{x})\right] \\
&= \frac{1}{2}\mathbf{E}_{\boldsymbol{x}\sim\mathcal{D}(\boldsymbol{x})}\mathbf{E}_{y\sim\mathcal{D}(y|\boldsymbol{x})}\left[\mathrm{tr}(\mathcal{J}_{\mathbf{w}_{\hat{\rho}}}(\boldsymbol{x})^\top\tilde{\mathbf{H}}^{-1}\mathcal{J}_{\mathbf{w}_{\hat{\rho}}}(\boldsymbol{x})\mathcal{J}_f(\boldsymbol{x})\mathcal{J}_f(\boldsymbol{x})^\top)\right] \\
&= \frac{1}{2}\mathbf{E}_{\boldsymbol{x}\sim\mathcal{D}(\boldsymbol{x})}\left[\mathrm{tr}(\mathcal{J}_{\mathbf{w}_{\hat{\rho}}}(\boldsymbol{x})^\top\tilde{\mathbf{H}}^{-1}\mathcal{J}_{\mathbf{w}_{\hat{\rho}}}(\boldsymbol{x})\mathbf{E}_{y\sim\mathcal{D}(y|\boldsymbol{x})}\left[\mathcal{J}_f(\boldsymbol{x})\mathcal{J}_f(\boldsymbol{x})^\top\right])\right] \\
&= \frac{1}{2}\mathbf{E}_{\boldsymbol{x}\sim\mathcal{D}(\boldsymbol{x})}\left[\mathrm{tr}(\mathcal{J}_{\mathbf{w}_{\hat{\rho}}}(\boldsymbol{x})^\top\tilde{\mathbf{H}}^{-1}\mathcal{J}_{\mathbf{w}_{\hat{\rho}}}(\boldsymbol{x})\mathbf{E}_{y\sim\mathcal{D}(y|\boldsymbol{x})}\left[\boldsymbol{y}\boldsymbol{y}^\top(\mathcal{J}_f(\boldsymbol{x})_y)^2\right])\right]
\end{aligned}
\tag{12}
$$

In the final line we have used both a one-hot encoding $\boldsymbol{y}$ and an integer encoding $y$ of the label. The substitution holds because $\mathcal{J}_f(\boldsymbol{x})_i$ is non zero only for $i = y$. In the above we note that 1) $\boldsymbol{y}\boldsymbol{y}^\top(\mathcal{J}_f(\boldsymbol{x})_y)^2$ is diagonal as the label $\boldsymbol{y}$ is a one-hot vector 2) For a model with finite weights, where $\|\mathbf{w}_{\hat{\rho}}\|_2^2$ is bounded, we can assume that $(\mathcal{J}_f(\boldsymbol{x})_y)^2 \geq c$, where $c$ is a positive constant. This is because the minimum for the cross-entropy loss is achieved at infinity for one of the input logits. We then get

$$
\begin{aligned}
\mathbf{V}(\hat{\rho}) &\gtrapprox \frac{1}{2}\mathbf{E}_{\boldsymbol{x}\sim\mathcal{D}(\boldsymbol{x})}\left[\mathrm{tr}(\mathcal{J}_{\mathbf{w}_{\hat{\rho}}}(\boldsymbol{x})^\top\tilde{\mathbf{H}}^{-1}\mathcal{J}_{\mathbf{w}_{\hat{\rho}}}(\boldsymbol{x})c\mathbf{I})\right] \\
&= \frac{c}{2}\mathbf{E}_{\boldsymbol{x}\sim\mathcal{D}(\boldsymbol{x})}\left[\mathrm{tr}(\mathcal{J}_{\mathbf{w}_{\hat{\rho}}}(\boldsymbol{x})^\top\tilde{\mathbf{H}}^{-1}\mathcal{J}_{\mathbf{w}_{\hat{\rho}}}(\boldsymbol{x}))\right].
\end{aligned}
\tag{13}
$$

$\square$

## 9   Deriving the diversity encouraging vector field

**Heuristic 1.** *For the Empirical Fisher $\mathbf{F}$ approximation to the Hessian, for a weight $\mathbf{w}_j$, and given training data $Z_\mathrm{t}$ and unlabeled data $Z_\mathrm{u}$, we propose the following diversity encouraging vector fields*

$$
\begin{aligned}
\Gamma_{\mathrm{SD}}(\mathbf{w})_j = &-2\lambda A_j(Z_\mathrm{t})^{-2}\left(\sum_{\boldsymbol{x}_\mathrm{t},y_\mathrm{t}\in Z_\mathrm{t}}\phi\left[\frac{\partial\log p(y_\mathrm{t};f)}{\partial\mathbf{w}_j}\right]^3\right)\sum_{\boldsymbol{x}_\mathrm{u}\in Z_\mathrm{u}}\sum_a\left(\frac{\partial f_a(\boldsymbol{x}_\mathrm{u};\mathbf{w})}{\partial\mathbf{w}_j}\right)^2 \\
&+ 2A_j(Z_\mathrm{t})^{-1}\sum_{\boldsymbol{x}_\mathrm{u}\in Z_\mathrm{u}}\sum_a\left(\chi\left[\frac{\partial f_a(\boldsymbol{x}_\mathrm{u},\mathbf{w})}{\partial\mathbf{w}_j}\right]^3\right),
\end{aligned}
$$

*and*

$$
\begin{aligned}
\Gamma_{\mathrm{SD+}}(\mathbf{w})_j = &-2\lambda A_j(Z_\mathrm{t})^{-2}\left(\sum_{\boldsymbol{x}_\mathrm{t},y_\mathrm{t}\in Z_\mathrm{t}}\phi\left[\frac{\partial\log p(y_\mathrm{t};f)}{\partial\mathbf{w}_j}\right]^3\right)\sum_{\boldsymbol{x}_\mathrm{u}\in Z_\mathrm{u}}\left(\frac{\partial f_{a_{\boldsymbol{x}}}(\boldsymbol{x}_\mathrm{u};\mathbf{w})}{\partial\mathbf{w}_j}\right)^2 \\
&+ 2A_j(Z_\mathrm{t})^{-1}\sum_{\boldsymbol{x}_\mathrm{u}\in Z_\mathrm{u}}\left(\chi\left[\frac{\partial f_{a_{\boldsymbol{x}}}(\boldsymbol{x}_\mathrm{u},\mathbf{w})}{\partial\mathbf{w}_j}\right]^3\right),
\end{aligned}
$$

*where $A_j(Z_\mathrm{t}) = \frac{1}{\sigma_\pi^2} + \lambda\sum_{\boldsymbol{x}_\mathrm{t},y_\mathrm{t}\in Z_\mathrm{t}}\left(\frac{\partial\log p(y_\mathrm{t};f)}{\partial\mathbf{w}_j}\right)^2$ and $\phi,\chi\in\mathbb{R}$.*

*Proof.* First note how naively taking the gradient of $\sum_{\boldsymbol{x}_u,y_u\in Z_\mathrm{u}}\sum_a\left[\mathcal{J}_{\mathbf{w}_{\hat{\rho}}}(\boldsymbol{x}_u)\tilde{\mathbf{H}}^{-1}\mathcal{J}_{\mathbf{w}_{\hat{\rho}}}(\boldsymbol{x}_u)^\top\right]_a$ without approximating $\mathbf{H}$ (which is part of $\tilde{\mathbf{H}}$) will result in third-order derivatives which are prohibitively expensive to compute. We thus propose to approximate $\mathbf{H}$ with the diagonal of the Empirical Fisher $\mathrm{diag}(\mathbf{F})$.

We write down the form of the diagonal elements of the Empirical Fisher $\mathbf{F}$ assuming also that we keep only the diagonal elements, and we only have one training sample $(\boldsymbol{x}_t, y_t)$. We get

$$
\mathbf{F}_a = \left(\frac{\partial\log p(y_t;f)}{\partial\mathbf{w}_a}\right)^2.
\tag{14}
$$

Since $\tilde{\mathbf{H}} \approx \lambda \mathrm{diag}(\mathbf{F}) + \frac{1}{\sigma_\pi^2}\mathbf{I}$, we get

$$\tilde{\mathbf{H}}_a \approx \lambda \left( \frac{\partial \log p(y_t; f)}{\partial \mathbf{w}_a} \right)^2 + \frac{1}{\sigma_\pi^2}. \tag{15}$$

Finally, the variance for each output of the linearized network and a single unlabeled sample $(\boldsymbol{x}_u, \cdot)$ is

$$\left[ \mathcal{J}_{\mathbf{w}_{\hat{\rho}}}(\boldsymbol{x}_u) \tilde{\mathbf{H}}^{-1} \mathcal{J}_{\mathbf{w}_{\hat{\rho}}}(\boldsymbol{x}_u)^\top \right]_a$$
$$\approx \sum_i \left[ \left( \lambda \left( \frac{\partial \log p(y_t; f)}{\partial \mathbf{w}_i} \right)^2 + \frac{1}{\sigma_\pi^2} \right)^{-1} \left( \frac{\partial f_a(\boldsymbol{x}_u; \mathbf{w})}{\partial \mathbf{w}_i} \right)^2 \right]. \tag{16}$$

We now sum over the different outputs $f_a$

$$\sum_a \left[ \mathcal{J}_{\mathbf{w}_{\hat{\rho}}}(\boldsymbol{x}_u) \tilde{\mathbf{H}}^{-1} \mathcal{J}_{\mathbf{w}_{\hat{\rho}}}(\boldsymbol{x}_u)^\top \right]_a$$
$$\approx \sum_a \sum_i \left[ \left( \lambda \left( \frac{\partial \log p(y_t; f)}{\partial \mathbf{w}_i} \right)^2 + \frac{1}{\sigma_\pi^2} \right)^{-1} \left( \frac{\partial f_a(\boldsymbol{x}_u; \mathbf{w})}{\partial \mathbf{w}_i} \right)^2 \right]. \tag{17}$$

We now take the partial derivative with respect to a weight $\mathbf{w}_j$. We get

$$\frac{\partial}{\partial \mathbf{w}_j} \sum_a \left[ \mathcal{J}_{\mathbf{w}_{\hat{\rho}}}(\boldsymbol{x}_u) \tilde{\mathbf{H}}^{-1} \mathcal{J}_{\mathbf{w}_{\hat{\rho}}}(\boldsymbol{x}_u)^\top \right]_a$$
$$\approx \frac{\partial}{\partial \mathbf{w}_j} \sum_i \left[ \left( \lambda \left( \frac{\partial \log p(y_t; f)}{\partial \mathbf{w}_i} \right)^2 + \frac{1}{\sigma_\pi^2} \right)^{-1} \sum_a \left( \frac{\partial f_a(\boldsymbol{x}_u; \mathbf{w})}{\partial \mathbf{w}_i} \right)^2 \right]$$
$$= -\sum_i \left( \lambda \left( \frac{\partial \log p(y_t; f)}{\partial \mathbf{w}_i} \right)^2 + \frac{1}{\sigma_\pi^2} \right)^{-2} \left( 2\lambda \left( \frac{\partial \log p(y_t; f)}{\partial \mathbf{w}_i} \frac{\partial^2 \log p(y_t; f)}{\partial \mathbf{w}_i \partial \mathbf{w}_j} \right) \right) \sum_a \left( \frac{\partial f_a(\boldsymbol{x}_u; \mathbf{w})}{\partial \mathbf{w}_i} \right)^2$$
$$+ \sum_i \left( \lambda \left( \frac{\partial \log p(y_t; f)}{\partial \mathbf{w}_i} \right)^2 + \frac{1}{\sigma_\pi^2} \right)^{-1} 2 \sum_a \left( \frac{\partial f_a(\boldsymbol{x}_u; \mathbf{w})}{\partial \mathbf{w}_i} \frac{\partial^2 f_a(\boldsymbol{x}_u; \mathbf{w})}{\partial \mathbf{w}_i \partial \mathbf{w}_j} \right). \tag{18}$$

We note how the above has the following general form

$$\Gamma(\mathbf{w})_j \triangleq -2\lambda \sum_i A_i(\boldsymbol{x}_t, y_t)^{-2} \left( U(\boldsymbol{x}_t; \mathbf{w})_{i,j} \frac{\partial \log p(y_t; f)}{\partial \mathbf{w}_i} \right) \sum_a \left( \frac{\partial f_a(\boldsymbol{x}_u; \mathbf{w})}{\partial \mathbf{w}_i} \right)^2$$
$$+ 2 \sum_i A_i(\boldsymbol{x}_t, y_t)^{-1} \sum_a \left( V_a(\boldsymbol{x}_u; \mathbf{w})_{i,j} \frac{\partial f_a(\boldsymbol{x}_u; \mathbf{w})}{\partial \mathbf{w}_i} \right), \tag{19}$$

where $A_i(\boldsymbol{x}_t, y_t) = \frac{1}{\sigma_\pi^2} + \lambda \left( \frac{\partial \log p(y_t; f)}{\partial \mathbf{w}_i} \right)^2$ and the matrices $U$ and $V_a$ are to be chosen. For the choices $U = \frac{\partial^2 \log p(y_t; f)}{\partial \mathbf{w}_i \partial \mathbf{w}_j}$ and $V_a = \frac{\partial^2 f_a(\boldsymbol{x}_u; \mathbf{w})}{\partial \mathbf{w}_i \partial \mathbf{w}_j}$ we recover the true gradient of the self-diversity term. However, we note that the final gradient is still expensive to compute.

Instead, we make the choice

$$U(\boldsymbol{x}_t; \mathbf{w})_{i,i} = \phi \left[ \frac{\partial \log p(y_t; f)}{\partial \mathbf{w}_i} \right]^2, \qquad V_a(\boldsymbol{x}_u; \mathbf{w})_{i,i} = \chi \left[ \frac{\partial f_a(\boldsymbol{x}_u; \mathbf{w})}{\partial \mathbf{w}_i} \right]^2, \tag{20}$$

and we restrict $\sum_i$ to $\sum_{i=j}$. This results in

$$\Gamma(\mathbf{w})_j = -2\lambda A_j(Z_t)^{-2} \phi \left[ \frac{\partial \log p(y_t; f)}{\partial \mathbf{w}_j} \right]^3 \sum_a \left( \frac{\partial f_a(\boldsymbol{x}_u; \mathbf{w})}{\partial \mathbf{w}_j} \right)^2$$
$$+ 2 A_j(Z_t)^{-1} \sum_a \chi \left[ \frac{\partial f_a(\boldsymbol{x}_u, \mathbf{w})}{\partial \mathbf{w}_j} \right]^3, \tag{21}$$

where $\phi, \chi \in \mathbb{R}$.

For an unlabeled dataset $Z_u$ and training dataset $Z_t$ we finally get

$$
\begin{aligned}
\Gamma(\mathbf{w})_j = -2\lambda \sum_{\boldsymbol{x}_t, y_t \in Z_t} \phi \left( \frac{\partial \log p(y_t; f)}{\partial \mathbf{w}_j} \right)^3 A(Z_t)^{-2} \sum_{\boldsymbol{x}_u \in Z_u} \sum_a \left( \frac{\partial f_a(\boldsymbol{x}_u; \mathbf{w})}{\partial \mathbf{w}_j} \right)^2 \\
+ A(Z_t)^{-1} 2 \sum_{\boldsymbol{x}_u \in Z_u} \sum_a \chi \left( \frac{\partial f_a(\boldsymbol{x}_u; f)}{\partial \mathbf{w}_j} \right)^3 .
\end{aligned}
\tag{22}
$$

where $A(Z_t) = \frac{1}{\sigma_\pi^2} + \lambda \sum_{\boldsymbol{x}_t, y_t \in Z_t} \left( \frac{\partial \log p(y_t; f)}{\partial \mathbf{w}_j} \right)^2$ and $\phi, \chi \in \mathbb{R}$. We use Equation (22) as the $\Gamma_{\mathrm{SO}}(\mathbf{w})$ diversity vector field, while $\Gamma_{\mathrm{SO+}}(\mathbf{w})$ results from restricting $\sum_a$ to $\sum_{a=a_{\boldsymbol{x}}}$.

$\square$

We now discuss our choices for $\chi$ and $\phi$. We plot in Figure 3 the final vector field $-(-\Gamma(\mathbf{w})_j)$ for $\lambda \gg 1$ and $1 \gg \sigma_\pi^2 > 0$ which are realistic ranges for the temperature and the prior, a single training sample $(\boldsymbol{x}_t, y_t)$, a single unlabeled sample $(\boldsymbol{x}_u, \cdot)$, a single weight $\mathbf{w}_j$, as well as a single output dimension $a$. We first notice that $\phi$ has a very small effect on the final gradient, inducing a slight asymmetry. In principle, we could also set $\phi = 0$ without any significant loss in our approximation. The most significant difference comes from the *sign* of $\chi$. For $\chi > 0$ the regulariser encourages our predictor to fit random labels on the unlabeled data. This is because $-(-\Gamma(\mathbf{w})_j)$ and $\frac{\partial f_a(\boldsymbol{x}_u; f)}{\partial \mathbf{w}_j}$ have the same sign. Note how $\frac{\partial f_a(\boldsymbol{x}_u; f)}{\partial \mathbf{w}_j}$ is the direction that maximizes the logit of class $a$. When $\chi < 0$ we get the opposite effect. We now encourage the probabilities for all classes (or a random class in the case of SO+) to be low for unlabeled data.

We ran small-scale experiments where choices other than $\phi = 1$ and $\chi = 1$ did not yield promising results. We thus selected $\phi = 1$ and $\chi = 1$ for our final regularizer.

**Motivating the choices $U(\boldsymbol{x}_t; \mathbf{w})_{i,i}$ and $V_a(\boldsymbol{x}_u; \mathbf{w})_{i,i}$**

We chose

$$
U(\boldsymbol{x}_t; \mathbf{w})_{i,i} = \phi \left[ \frac{\partial \log p(y_t; f)}{\partial \mathbf{w}_i} \right]^2, \qquad V_a(\boldsymbol{x}_u; \mathbf{w})_{i,i} = \chi \left[ \frac{\partial f_a(\boldsymbol{x}_u; \mathbf{w})}{\partial \mathbf{w}_i} \right]^2 .
\tag{23}
$$

Consider the nonlinear least squares problem $\mathcal{L}(\mathbf{w}) = \frac{1}{2}(f(\boldsymbol{x}; \mathbf{w}) - y)^2$ then the Hessian can be written as

$$
\nabla^2 \mathcal{L}(\mathbf{w}) = \nabla_{\mathbf{w}} f(\boldsymbol{x}; \mathbf{w}) \nabla_{\mathbf{w}} f(\boldsymbol{x}; \mathbf{w})^\top + r \nabla_{\mathbf{w}}^2 f(\boldsymbol{x}; \mathbf{w})
$$

where $r = f(\boldsymbol{x}; \mathbf{w}) - y$ is the residual. As such for the square loss both $\nabla_{\mathbf{w}} f(\boldsymbol{x}; \mathbf{w}) \nabla_{\mathbf{w}} f(\boldsymbol{x}; \mathbf{w})^\top$ and $\nabla_{\mathbf{w}}^2 f(\boldsymbol{x}; \mathbf{w})$ provide some information on the curvature. This is our motivation for using $\nabla_{\mathbf{w}} f(\boldsymbol{x}; \mathbf{w}) \nabla_{\mathbf{w}} f(\boldsymbol{x}; \mathbf{w})^\top$ as a source of curvature information in the place of $\nabla_{\mathbf{w}}^2 f(\boldsymbol{x}; \mathbf{w})$. This resulted in exploring $V_a(\boldsymbol{x}_u; \mathbf{w})_{i,i} = \chi \left[ \frac{\partial f_a(\boldsymbol{x}_u; \mathbf{w})}{\partial \mathbf{w}_i} \right]^2$.

Similarly, it is interesting to explore $U(\boldsymbol{x}_t; \mathbf{w})_{i,i} = \phi \left[ \frac{\partial \log p(y_t; f)}{\partial \mathbf{w}_i} \right]^2$ as we have already approximated second-order derivates of the loss with squares of the first-order derivatives when approximating the Hessian with the Fisher. Both of the above are heuristics, and as we are forcing the corresponding matrix entries to have a positive sign we explored $\chi, \phi \in \mathbb{R}$ which change this sign.

## 10  Pareto optimality

In multi-objective optimization, the Pareto front (also called Pareto frontier or Pareto curve) is the set of all Pareto efficient solutions. Consider $A$ a set of criterion vectors in $\mathbb{R}^m$. Assume that the preferred directions of criteria values is known. A point $a'' \in \mathbb{R}^m$ is preferred to (strictly dominates) another point $a' \in \mathbb{R}^m$, written as $a'' \succ a'$, when $a''$ improves all available criteria jointly compared to $a'$. The Pareto frontier is thus written as:

$$
P(A) = \{ a' \in A : \{ a'' \in A : a'' \succ a', a' \neq a'' \} = \emptyset \}.
$$

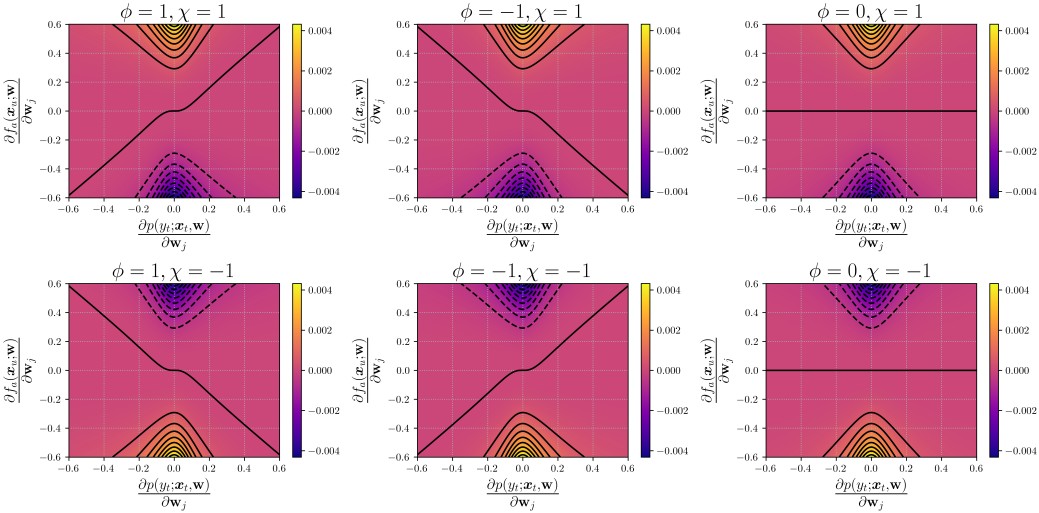

Figure 3: We plot $-(-\Gamma(\mathbf{w})_j)$ with $\phi, \chi \in \{-1, 0, 1\}$ for $\lambda \gg 1$ and $1 \gg \sigma_\pi^2 > 0$ which are realistic ranges for the temperature and the prior, a single training sample $(\boldsymbol{x}_{\mathrm{t}}, y_{\mathrm{t}})$, a single unlabeled sample $(\boldsymbol{x}_{\mathrm{u}}, \cdot)$, a single weight $\mathbf{w}_j$, as well as a single output dimension $a$. We first notice that $\phi$ has a very small effect on the final gradient, inducing a slight asymmetry. The most significant difference comes from the *sign* of $\chi$. For $\chi > 0$ the regulariser encourages our predictor to fit random labels on the unlabeled data. This is because $-(-\Gamma(\mathbf{w})_j)$ and $\frac{\partial f_a(\boldsymbol{x}_j; f)}{\partial \mathbf{w}_j}$ have the same sign (where we either select $a$ at random in SO+ or we optimize over all $a$ in SO). Note how $\frac{\partial f_a(\boldsymbol{x}_j; f)}{\partial \mathbf{w}_j}$ is the direction that maximizes the logit of class $a$. When $\chi < 0$ we get the opposite effect. We now encourage the logits for all classes (or a random class) to be low for unlabeled data.

The ideal point is the point that optimizes all criteria in the best possible way. In our case we use $a = [1 - \mathrm{Acc}, \mathrm{TACE}]$ as our criterion vector with $a \in (0, 1)^2$. Our ideal point is then $a^\star = [0, 0]$ the point where both the misclassification rate and the TACE are 0. Intuitively this predictor makes both perfect and perfectly calibrated predictions. In the absence of other criteria, the point closest to the ideal point $\arg\min_{a \in P(A)} \|a - a^\star\|_2^2$ is often considered the optimal one. We use the phrase "Pareto optimal" in this sense. Note that alternatively one can refer to all points on the Pareto front as optimal with the point closest to the ideal point referred to as the knee point.

## 11   Experimental setup

We run our experiments on a combination of NVIDIA A100 and V100 GPUs, on our local cluster. The total computation time, including hyperparameter tuning and training, was approximately 1600 GPU hours. Hyperparameter tuning was done over a single random seed per each ensemble member due to the computational cost.

In the following list, we include the libraries and datasets that we used together with their corresponding licenses

- PyTorch package [Paszke et al., 2019]: Modified BSD Licence
- MNIST-10 dataset [Deng, 2012]: MIT Licence
- CIFAR-10 dataset [Krizhevsky and Hinton, 2009]: MIT Licence
- CIFAR-100 dataset [Krizhevsky and Hinton, 2009]: MIT Licence
- SVHN dataset [Netzer et al., 2011]: -
- FashionMnist dataset [Xiao et al., 2017]: MIT Licence
- JAX [Bradbury et al., 2018]: Apache License 2.0
- flax [Heek et al., 2023]: Apache License 2.0

## 12 More experiments

We include here experiments on the CIFAR-10 and CIFAR-100 datasets, for the MLP and LeNet architectures and the case of data augmentation. We use random flips and crops as is the standard for these two datasets. We observe similar results to the main text. We achieve the best test accuracy in all experiments. At the same time we are Pareto optimal in 3 out of 4 cases.

**Computational and Memory Cost** Our memory cost increases linearly with the number of classes for the SO algorithm. For the SO+ algorithm, the additional memory cost is constant. At the same time, the computation time for the SO algorithm roughly increased roughly by a factor of 2. For the SO+ algorithm, we sometimes observed an increase by a factor of 10. This is potentially because we need to estimate per sample *and per output* gradients potentially degrading parallelization.

## 13 Effect of Ensemble Size

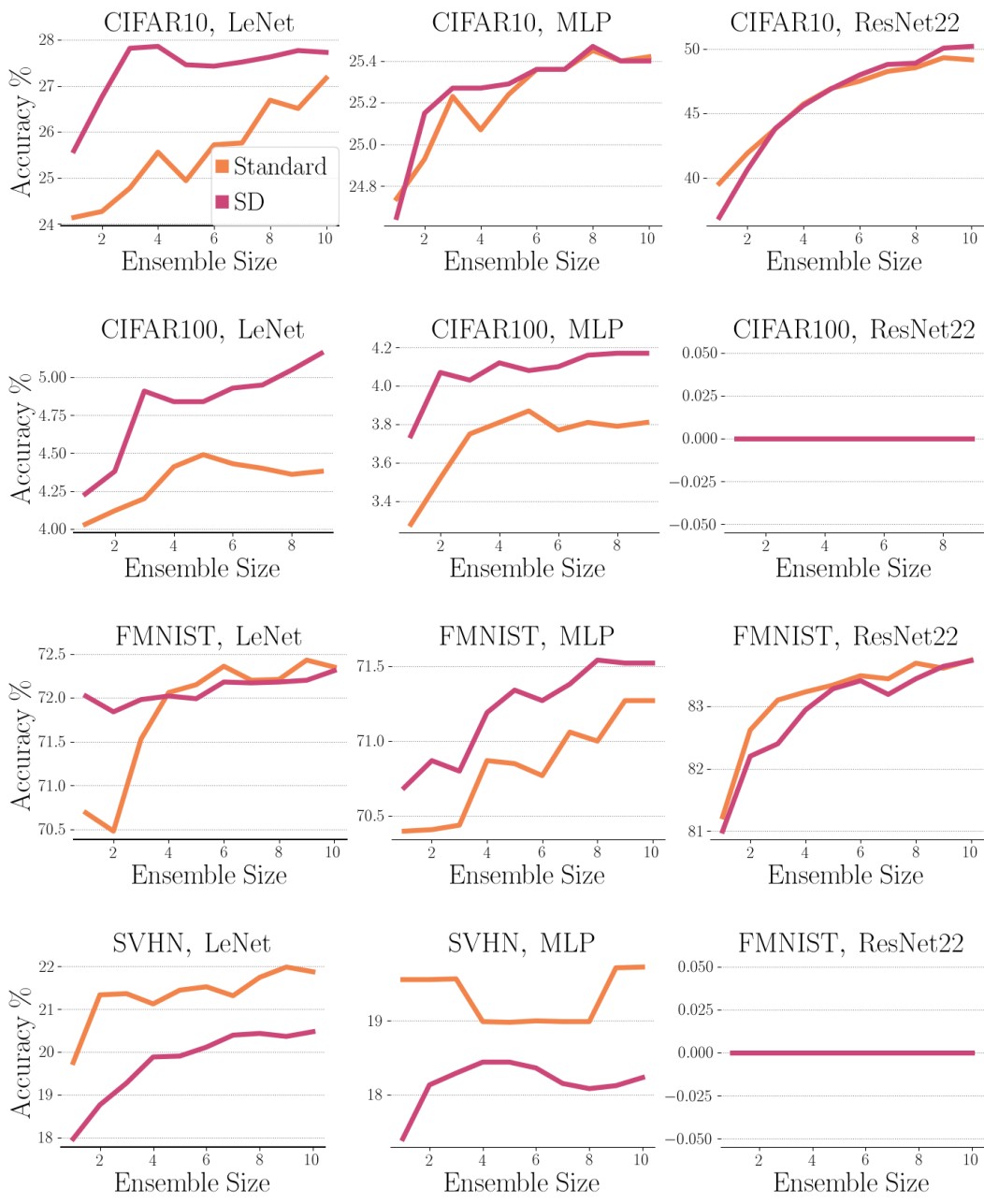

Figure 4: The effect of the ensemble size on CIFAR10/ CIFAR100/ FMNIST/ SVHN and LeNet/ MLP/ ResNet22 in the small data regime. In most cases SO ensembles achieve better test accuracy with fewer ensemble members than Standard ensembles.

