# OpenReview forum: "Improving Deep Ensembles without Communication"
_NeurIPS.cc/2023/Workshop/WANT — WANT@NeurIPS 2023 Poster_

### Official Review · Reviewer_tRS1 · 2023-10-24
**Strong paper with some flaws**

**Confidence:** 4

**Review:**

The paper describes a novel (and often SOTA in some benchmarks) method of deep ensemble called SO/SO+. The method minimizes a tighter upper bound on Bayesian risk and adds a diversity term to the classification loss. This diversity term requires computations of the network Jacobian in order to increase variance into weights. This also improves computation efficiency, as the ensemble members do not need to communicate with each other.

Strengths:
- It is hard to find significant mistakes in formulations and proofs of the theorems. The notation is mostly clear.
- The evaluation results on the benchmarks provided seem to be legit and correct. Numbers look realistic -- and it's indeed a SOTA on the tasks discussed.

Weaknesses:
- In section 3.4, there's no evidence that this computation of the approximate Hessian is faster than the precise one. There's a sum of as many terms as number of tensors in the network, matrix multiplications of the tensor sizes and first order derivatives w.r.t to the tensors which seem to have complexity of O(N^4), precise Hessian computation may be as expensive
- From section 3.4, it is not clear how tight this approximation of the Hessian
- There's no comparison with the non-ensemble WideResNet/LetNet/MLP.
- The paper does not have any comparison in terms of savings on communication -- this was one of the main points of the proposed method, That would be great to have some kind of description of the potential benefits

Questions and further discussion:
- Why WideResNet and not more efficient and novel architectures (EfficientNet or whatever)? That would've saved a lot of time as they are more efficient and also benchmarking results on the discussed datasets are well known for such architectures

Conclusion:
I still believe that the results are precise. I give this paper 4

---

### Official Review · Reviewer_G7LA · 2023-10-24
**Review-Ensembles without communication**

**Confidence:** 4

**Review:**

#### **Summary and contributions:**
The paper tackles the problem of inducing diversity in ensemble members. It builds up on the second-order PAC Bayes bound approach proposed by Masegosa (NeurIPS 2020), where instead of jointly optimizing the ensemble member parameters, where the empirical variance term in the second-order PAC Bayes bound induces generalization (as per Masegosa paper), this paper trains ensemble members independently without parameter sharing, intending to diversify the ensemble solution further. They introduce two optimization strategies: second-order (SO) and randomized second-order (SO+). In the SO strategy, the full Jacobian is computed for each unlabeled sample (associated with a linear increase in memory costs with the number of classes). In contrast, the SO+ strategy computes the gradients with respect to the network output without having to incur additional costs associated with the full Jacobian computation.

#### **Strengths:**
Extensive experimentation with toy data and open datasets (CIFAR-10, CIFAR-100, fashion MNIST, SVHN). Performance comparisons were made with several other approaches for inducing diversity in ensembles, standard ensembles, second-order PAC Bayes (Masegosa NeurIPS 2020), and the more recently published method D-BAT by Pagliardini, et al. ICLR 2023, where ensembles are pushed to have similar performance on training distribution and varying performance on out-of-distribution data for improving generalizability. Performance was measured with several metrics (accuracy, ECE, TACE, Brier and NLL). The objective was tested with several model architectures: ResNet22, LeNet, MLP.

#### **Weaknesses:**
The paper lacks clarity, and there's a lack of flow between the sections, especially in the methodology section, making it difficult to read and follow what the authors are trying to convey.

Specific comments on certain statements made in the paper:
- lines 22-23: "...biasing the minima to have desirable generalization properties." It is unclear what the authors are referring to with the phrase "biasing the minima".
- lines 28-29: confusing sentence: "enforcing diversity intuitively requires computing some mean predictions with respect to which the ensemble members are pushed away", difficult to understand what the authors are trying to say in this sentence.
- lines 40-41: "...optimize a novel PAC-Bayes bound..." how is it different than the PAC-Bayes proposed by Masegosa NeurIPS 2020?
- line 41: "..., we use a different seed for each ensemble member to increase diversity.", this approach is a given when working with ensembles to ensure variation in the random initialization points for the member parameters, and variation in the random shuffling of the training data.
- line 54: "However, the ensemble size increases quadratically." If I understand what the authors are trying to convey in this sentence is referring to the increase in the number of parameters required for the ensemble, and if that is the message, then this sentence is false, as with ensembles, each additional member will add N- parameters, and with k-members, then the size of the ensemble is of kN  <<< N\^2 -- as usually, the number of ensemble members is below 20.
- lines 81-87: This paragraph is extremely convoluted and unclear, it is hard to discern what the authors are trying to convey. What are the authors referring to with the "stochastic objective", what does "enforce the mean and the covariance to have some desired values in the appropriate regions of the input space" mean?
- line 106: What does "Out-of-sample" mean? Is this referring to out-of-distribution? Please clarify.
- Figure 4: subfigures for CIFAR100 and FashionMNIST ResNet22 have no data?

There's also a weak/unclear aim of the paper to the theme of the WANT workshop. What is the computational gain/loss for this approach? The SO strategy would incur significant computational costs due to the Jacobian calculation for each unlabeled sample. The authors also state in line 438 that the SO+ strategy can increase computational time by a factor of 10. Therefore, it seems like these strategies would incur significant computational costs without significant gains in performance, as summarized in Tables 1 and 2. Tables 1 and 2 and Figure 2 show little difference in performance when comparing SO+ strategy to the standard simple ensemble approach. At the same time, there's inconsistent gain (sometimes loss) in performance with respect to accuracy, uncertainty, and calibration. If the paper aims to target parallelization since these strategies do not require joint training of the ensemble members, then a clear statement on that regard and training scheme should be explained/included. Also, if the goal is to improve the parallel training of ensemble members, the benefits of the proposed strategies are lacking over the standard ensemble.

#### **Rating:**
4: An okay submission, but not good enough; a reject.

---

### Official Review · Reviewer_5c84 · 2023-10-24
**Accept**

**Confidence:** 4

**Review:**

They introduce a communication-free algorithm for deep ensembles, optimizing a more stringent PAC-Bayesian bound. Their writing is clear and effective. The method is well-supported by theoretical foundations, and they perform experiments on various datasets ranging from MNIST to CIFAR, employing diverse model architectures and providing adequate ablation studies.

---

### Meta-Review · Area_Chair_LvM3 · 2023-10-27

**Recommendation:** Accept (Poster)
**Confidence:** 4

**Metareview:**

There is a lot of valid critiscim for this paper by one of the reviewers and I suggest the authors take it into account seriously. From my perspective, I have a major concerns about the main Theorem, which is written in a fairly sloppy way (even the formulations are not mathematically strict allowing for terms like "up to an approximation"). The proofs are also quite sloppy. I suggest the authors to work on the proofs and formulations to have it clean for future submissions. Still I think that the paper can be presented at the workshop as some of the ideas are quite nice.

---

### Decision · Program_Chairs · 2023-10-28

**Decision:**

Accept (Poster)

**Comment:**

We thank the authors for their time and contribution to WANT and we are pleased to share that after the reviewing process the paper has been accepted. Congratulations! We encourage the authors to consider reviewers' feedback for the improvement of the camera-ready version. We hope to see you in person at the workshop and brainstorm on efficient training research together!